# A Comparative Review of Cytokines and Cytokine Targeting in Sepsis: From Humans to Horses

**DOI:** 10.3390/cells13171489

**Published:** 2024-09-05

**Authors:** Kallie J. Hobbs, Rosemary Bayless, M. Katie Sheats

**Affiliations:** 1Department of Clinical Sciences, North Carolina State University, Raleigh, NC 27526, USA; kjhobbs@ncsu.edu; 2Department of Molecular Biomedical Sciences, North Carolina State University, Raleigh, NC 27526, USA; rlbayles@ncsu.edu

**Keywords:** cytokine, sepsis, equine, hemoperfusion

## Abstract

With the emergence of COVID-19, there is an increased focus in human literature on cytokine production, the implications of cytokine overproduction, and the development of novel cytokine-targeting therapies for use during sepsis. In addition to viral infections such as COVID-19, bacterial infections resulting in exposure to endotoxins and exotoxins in humans can also lead to sepsis, resulting in organ failure and death. Like humans, horses are exquisitely sensitive to endotoxin and are among the veterinary species that develop clinical sepsis similar to humans. These similarities suggest that horses may serve as a naturally occurring model of human sepsis. Indeed, evidence shows that both species experience cytokine dysregulation, severe neutropenia, the formation of neutrophil extracellular traps, and decreased perfusion parameters during sepsis. Sepsis treatments that target cytokines in both species include hemoperfusion therapy, steroids, antioxidants, and immunomodulation therapy. This review will present the shared cytokine physiology across humans and horses as well as historical and updated perspectives on cytokine-targeting therapy. Finally, this review will discuss the potential benefits of increased knowledge of equine cytokine mechanisms and their potential positive impact on human medicine.

## 1. Introduction

In humans, infections resulting in exposure to endotoxins and exotoxins can lead to sepsis, resulting in organ failure and death [1]. Horses and humans are both exquisitely sensitive to endotoxins, and like humans, horses also develop sepsis that can be life-threatening. Growing evidence supports that horses may exhibit a similar immune response to humans undergoing sepsis [1] as both species experience cytokine storms, severe neutropenia, the formation of neutrophil extracellular traps, and decreased perfusion parameters during sepsis [2]. These shared mechanisms of sepsis support horses as a comparative and translational model for human sepsis [1]. In both humans and horses, adult and neonatal sepsis is associated with high costs and high mortality rates. Between 2004 and 2018, the human adult sepsis mortality rate per 1,000,000 population was: 111.8 from pulmonary sepsis, 46.7 from abdominal sepsis, and 52 from genitourinary sepsis [3]. This is in contrast to neonatal humans, in which mortality was reported to be as high as 3930 per 100,000 births in the period 2009–2018 [4]. In adult horses, mortality from sepsis ranges from 37–79% depending on the inciting cause [5], and neonatal sepsis is a leading cause of referral to tertiary referral centers and mortality in foals [6].

In human medicine, sepsis is defined as a life-threatening organ dysfunction caused by a dysregulated host response to infection [7]. This can be triggered by the introduction of endotoxin from gram-negative bacteria or exotoxin from gram-positive bacteria entering the bloodstream. In equine medicine, the consensus on what defines sepsis is still a topic of debate, as it is often difficult to show definitive evidence of infection, even during states of severe systemic inflammatory response (SIRS) [8]. Though there is currently no consensus definition for sepsis in equine patients, several studies have extrapolated the SIRS definition from human medicine using parameters that include hyper- or hypothermia, tachycardia, tachypnea, and abnormal leukocyte count to identify horses with SIRS/sepsis [6]. While there is some agreement among equine clinicians and researchers on SIRS criteria, there is still a lack of consensus on definitions of sepsis and organ dysfunction. This is despite recent work by Roy et al., which investigated the association between SIRS parameters and outcomes in 479 adult horses presented on emergency [9]. Importantly, this work showed that the highest odds of death were in horses with all four SIRS criteria (SIRS4) and markers of altered tissue perfusion (i.e., blood lactate and altered mucus membrane color). The results of this study offer clinically relevant information and outcomes data that show the importance of altered tissue perfusion in the pathophysiology of life-threatening SIRS (i.e., sepsis). Equine veterinary clinicians and researchers need to build on these findings to develop consensus criteria for sepsis in equine patients.

Historically, sepsis has been viewed as an excessive systemic proinflammatory reaction to invasive microbial pathogens. In the recent literature, it has been proposed that the early phase of hyperinflammation is followed by a prolonged state of immunosuppression, termed immune paralysis. Immune paralysis results in impaired innate and adaptive immune responses, resulting in tissue damage and organ failure during sepsis [9]. Cytokines are the primary regulators of a variety of these inflammatory and anti-inflammatory responses.

In the recent literature, for both species, there is an increased focus on the mechanisms of cytokine production and the consequences of cytokine overproduction during sepsis [10,11]. Improved understanding of these mechanisms within the biomedical community is leading to increased interest in the development of novel cytokine-targeting therapies such as cytokine removal and immunomodulation treatments for use during sepsis. In a review of the literature indexed in PubMed from 1 August 2019 to 1 August 2024, 1513 manuscripts demonstrate linked medical treatments between humans and horses, supporting their shared pathology of sepsis. To date, there is no single treatment that has been identified as successful for resolving all cases of sepsis. As a result, further investigations into the roles of cytokines in sepsis and the therapeutic potential of cytokine modulation techniques are needed. Mechanisms discussed in this review may provide the groundwork for future research into cytokine modulation in both species.

## 2. Pathophysiology of Sepsis and Cytokines Roles

### 2.1. Cytokines

Cytokines are small soluble proteins secreted by a wide variety of cells, including immune cells (e.g., macrophages, neutrophils, dendritic cells, and T and B lymphocytes) and non-immune cells (e.g., endothelium, epithelium, and fibroblasts). These molecules play a pivotal role in cellular signaling and immune system activation [12,13]. In sepsis, these molecules undergo extreme dysregulation leading to deleterious effects such as dysregulation of apoptosis, increases in reactive oxygen species, and alternated signaling pathways of immune cells (Figure 1) [14]. There is supportive evidence from both clinical trials and literature meta-analyses in both species that increased plasma levels of cytokines in sepsis have a significant correlation with disease severity and patient survival (Table 1) [15,16].

Pro-inflammatory cytokines are a group of signaling molecules that drive the body’s immune response. Pro-inflammatory cytokines serve as key mediators in the initiation and regulation of inflammation, which is a fundamental component of the immune response to infection, injury, or other insults [17]. Sepsis can lead to the excessive production of pro-inflammatory cytokines such as interleukin-6 (IL-6), interleukin-1 beta (IL-1β), and tumor necrosis factor-alpha (TNF-α). This overproduction leads to downstream effects such as fever, tissue necrosis, and multisystemic organ failure [18].

Anti-inflammatory cytokines are a group of signaling molecules that play a crucial role in regulating the immune response and maintaining immune homeostasis. Unlike pro-inflammatory cytokines, which promote inflammation and immune activation, anti-inflammatory cytokines exert immunosuppressive effects and help to dampen excessive immune responses. These cytokines are involved in modulating the intensity and duration of inflammation, promoting tissue repair, and preventing immune-mediated damage. Anti-inflammatory cytokines are produced by various immune cells, including regulatory T cells (Tregs), type 2 T helper cells (Th2), and M2 macrophages, as well as by non-immune cells such as epithelial cells and fibroblasts [19]. Their production is induced in response to inflammatory stimuli and serves as a negative feedback mechanism to regulate immune responses. In both humans and horses, IL-10 is the most investigated anti-inflammatory cytokine.[20]

This review will focus on IL-6, IL-1β, TNF-α, and IL-10 (Table 2) due to their well understood pathology in sepsis to further explore the shared pathophysiology of sepsis across both humans and horses.

### 2.2. IL-6

IL-6 is a major mediator of the acute-phase response, which is a systemic reaction to infection, inflammation, or tissue injury. IL-6 is released in response to IL-1β and TNF-α and stimulates the production of acute-phase proteins, such as C-reactive protein (CRP) and fibrinogen, which play roles in host defense, tissue repair, and modulation of the immune response [21,22]. Anti-inflammatory properties of IL-6 have also been noted, including T-cell differentiation and inflammatory cytokine production [23]. Viva et al. and Molano et al. reported that plasma levels of IL-6 in human patients with sepsis are elevated in the early course of disease and are associated with disease severity, organ dysfunction, and mortality [24,25]. In horses, the production of IL-6 also occurs in response to infection, has been shown to be significantly elevated in cases of sepsis or severe SIRS, and has been correlated with disease severity and outcome [26]. For example, horses with abdominal pain that have higher peritoneal fluid IL-6 concentrations at admission are more likely to develop organ-damage, such as laminitis, or to be euthanized due to poor prognosis [27]. Additionally, increased serum concentrations of IL-6 in horses have been noted in mares with placentitis [28] and in horses with equine metabolic syndrome [29].

### 2.3. IL-1β

IL-1β is a potent inducer of inflammation and immune activation and a primary mediator of SIRS. IL-1β is produced by activated macrophages and is involved in many cellular activities, including cell proliferation, differentiation, and apoptosis. IL-1β can have beneficial effects for the host during times of stress, but it can also contribute to morbidity and mortality if produced in excessive quantities or for extended periods of time. Gou et al. found that in mice, IL-1β triggered the redistribution of CD11c^-^CD45RB^high^ dendritic cells (DC) as well as bone marrow cells (BMC) in parabiosis models. These findings suggest that IL-1β protects against sepsis by stimulating the local proliferation and differentiation of BMCs into CD11c^-^CD45RB^high^ DCs at immune organs and non-immune organs during sepsis [30]. Lemon et al. found that mice administered intranasal IL-1β demonstrated improved clearance of Streptococcus pneumonia, which suggests a role for IL-1β in macrophage recruitment and the clearance of Streptococcus pneumoniae [31]. In humans, serum IL-1β levels are persistently increased in patients with sepsis who have outcomes of non-survival, suggesting that it may also play a role in sepsis. Additionally, polymorphisms of IL-1β have been linked to the development of septic shock and death in human patients and may influence the immune response to major trauma [32,33]. In horses, serum IL-1β elevations have been documented in adult horses following experimentally induced endotoxemia [34]. IL-1β gene expression was significantly higher in foals with naturally-occurring sepsis than foals with neonatal maladjustment syndrome. In this population of septic foals, IL-1β expression was correlated to the neutrophil-to-lymphocyte ratio and monocyte count [35].

### 2.4. TNF-α

TNF-α exerts pleiotropic effects on various cell types and tissues. It stimulates the production of other pro-inflammatory cytokines, such as IL-1β and IL-6, and promotes the expression of adhesion molecules on endothelial cells, facilitating leukocyte migration and infiltration into inflamed tissues [36]. TNF-α is involved in the regulation of apoptosis and can induce cell death in certain cell types, particularly tumor cells and infected cells [12]. Dysregulated production of TNF-α is implicated in the pathogenesis of various inflammatory and autoimmune diseases in both humans and horses, leading to multiple organ failure [37]. Gharamti et al. reported elevated levels of TNF-α in blood were associated with increased mortality over a 28-day period in patients admitted to the ICU with sepsis [38]. In contrast, Rigato et al. described a correlation between low TNF-α production and increased mortality in patients with sepsis, suggesting that TNF-α may play a key role in human patients’ response to sepsis [39]. In horses, there is evidence that TNF-α may lead to laminitis, which is one of the “organs” that can be damaged during equine sepsis [40]. These findings are consistent with growing evidence that TNF-α production in sepsis may create vascular injury by promoting inappropriate vasodilation and vasoconstriction and contributing to other organ injury, including the development of acute kidney injury in horses with sepsis [41]. Additionally, Moore et al. found that increased levels of TNF-α are associated with poor outcomes in cases of equine colic, though these cases were not evaluated for sepsis as a complicating factor [42].

### 2.5. IL-10

IL-10 is a potent anti-inflammatory molecule. The effects of IL-10 include the down-regulation of key signaling receptors (e.g., CD40, CD80, CD86, and MHC II) on antigen presenting cells, the inhibition of neutrophil oxidative burst, the suppression of T cell proliferation, and the suppression of natural killer cell function [43]. In human patients with sepsis, an IL-10 blockade has been shown to decrease survival and increase neutrophil activation [44]. L’Heureux et al. found that in patients with sepsis-induced acute respiratory distress syndrome, plasma IL-10 concentrations were higher in non-survivors compared to survivors [45]. Similarly, septic neonatal foals that did not survive had significantly elevated levels of IL-10 compared to survivors [46]. In adult horses with experimentally induced endotoxemia, IL-10 gene expression in whole blood was not higher at any timepoint in comparison with the baseline [34]. This contrast in findings between neonatal and adult horses highlights current gaps in understanding regarding the role of IL-10 in the pathology of endotoxemia and sepsis in horses.

## 3. Cytokine Targeting as a Treatment for Sepsis

Currently, sepsis has no proven pharmacologic treatment other than supportive care (e.g., antibiotics, cardiovascular support, and corticosteroid administration) [47]. Because cytokines play a significant role in the dysregulated inflammatory response in sepsis patients, cytokines are an attractive target for novel sepsis therapies. Strategies for cytokine-targeted therapies can be classified under various domains: anti-inflammatory, anti-endotoxic, or immunomodulatory. Both anti-inflammatory and anti-endotoxic therapies can have undesired effects, such as kidney injury, immune suppression, and epithelial barrier dysfunction. As a result of the potentially deleterious effects of anti-inflammatory and anti-endotoxic therapies, immune modulation for the treatment of cytokine dysregulation in sepsis has become a cornerstone of recent research.

The US National Library of Medicine records of clinical trials related to sepsis and cytokines show a sharp increase in the past five years. In total, there are 293 registered clinical trials for cytokine modulation therapy. When these clinical trials are filtered to the last 10 years, 169 of these clinical trials started between 1 August 2014 and 1 August 2024. Cytokine removal therapy via hemoperfusion therapy is the central focus of 29 of these clinical trials. A search of the current literature using the PubMed database with the terms “sepsis”, “cytokines”, and dates 1 August 2019–1 August 2024 yields 88,485 manuscripts, of which 12,028 list cytokine targeting as a primary mechanism of treatment. When this database was evaluated further, most immunomodulatory treatments for sepsis fell into one of three categories: monoclonal antibody treatment, cytokine receptor blockade, or cytokine removal.

### 3.1. Monoclonal Antibody Treatment

The first use of monoclonal antibodies (mAb) was in 1896, when they were approved for use in preventing the rejection of transplanted kidneys. Since that time, over 30 mAb have been approved for human use in sepsis and sepsis-related conditions [48] Monoclonal antibodies are specifically designed to target against a specific epitope of an antigen and recently have been bound to a linker for targeted therapeutic medication delivery [49]. In an equine model of experimentally induced endotoxemia, miniature horses treated with mAb against TNF-α had significantly lower plasma TNF-α activity compared with controls. The TNF-α mAb group also had significantly lower clinical abnormality scores, lower heart rates, and higher WBC counts compared with miniature horses administered isotype control mAb [50]. Treatment with anti-TNF-α mAb therapy in miniature horses receiving an LPS infusion also modulated the production of IL-6, lactate, and prostacyclin [51]. More recently, anti-IL-5 mAbs have been explored as an adjunctive treatment in horses with insect bite hypersensitivity [52]. In human COVID-19 patients, mAbs against cytokines or cytokine-receptors reduced hospitalization and the need for mechanical ventilation [48]. It is important to note though that in 2022 the FDA banned many mAbs for use in COVID-19 due to concerns about adverse side effects, emphasizing the potential harmful consequences of blocking normal host responses to infections.

### 3.2. Cytokine Antagonist

Cytokine antagonists can inhibit the action of cytokines by acting directly on receptors to create a long-term immune blockade, by affecting the production of cytokines, or by binding to cytokines and preventing their subsequent action [53]. In humans with sepsis, the blockade of the IL-6 receptor reduced incidences of sepsis-related mortality and reduced instances of sepsis-related critical care admission [54]. Additionally, the blockade of the CXCR1 receptor has been shown to decrease lung injury, thrombosis, and the production of neutrophil extracellular traps in human and murine models of sepsis [55]. In horses, cytokine antagonist therapy is currently only documented for osteoarthritis using an IL-1 receptor antagonist [56]. The reduction of inflammatory pathways when using the IL-1 receptor antagonist in horses may indicate that further exploration for use in sepsis is warranted.

Although cytokine antagonists are generally more cost-effective compared to mAbs against cytokines, cytokine antagonist therapies can have similar undesirable effects associated with a decreased host response to infection.

### 3.3. Cytokine Removal

Beyond blocking cytokines and/or cytokine receptors with mAb or antagonist medications, there is increasing interest in removing circulating cytokines from septic patients via extracorporeal hemoperfusion therapy [57]. Hemoperfusion therapy uses a peristaltic pump to circulate anticoagulated blood from a patient through a column containing filter medium (hemadsorption column) and then return the blood back to the patient (Figure 2). Hemoperfusion therapy provides human physicians with a more controlled approach to targeting cytokines in septic patients, compared to other treatment approaches such as mAbs or cytokine antagonists that can cause a long-term immune blockade [58]. There is also growing evidence that hemoperfusion may reduce direct tissue damage from leukocytes by removing circulating neutrophil extracellular traps and the inhibition of neutrophil-reactive oxygen species [59]. Hemoperfusion therapy devices rely on one of three mechanisms: nonselective binding to beads, selective binding to beads, or binding to a permeable membrane.

In nonselective binding, molecules of sizes 5–60 kd are removed by the filter. This size range includes substances such as cytokines, bilirubin, myoglobin, protein-bound medications, exotoxin, and some synthetic toxins (e.g., rodenticide). These nonselective devices typically either have polymer beads, such as those used in Cytosorb, or activated-carbon beads, such as the Baxter HA330. The nonselective polymer-based devices use a concentration-dependent gradient to remove inflammatory cytokines. With this mechanism, high plasma concentrations of cytokines are cleared more efficiently than lower concentrations, reducing the potential negative consequences of the complete removal of cytokines. Complete removal could be detrimental to patient health, as cytokines in physiologic concentrations and under proper regulation can play a protective role in the immune system [11]. Hemoperfusion using a polymer-based column (CytoSorb) has been shown to decrease cytokine concentrations in humans with experimentally induced sepsis [60]. This same polymer-based column decreased mortality and vasoplegic shock in patients receiving extracorporeal life support [61] and decreased mechanical ventilation in COVID-19 patients [62]. In contrast, nonselective activated carbon devices do not provide gradient removal and instead work through a mesoporous system that adsorbs cytokines. While an activated carbon filter had minimal effects on cytokine concentrations in a human clinical trial, the device significantly reduced the amount of endotoxin in blood following hemoperfusion [63]. In horses, a polymer-based device (VetResQ) has been confirmed to remove cytokines ex vivo in LPS-treated equine blood [64] and has been shown to be safe to use in adult horses in both experimental and clinical conditions [65,66]. Additional experimental trials using the VetResQ device in horses receiving LPS infusions are ongoing. Selective binding filtration systems use polymer beads that are coated with an attractant. One of the most well-known of these devices in human medicine is Toraymyxin, which uses polymyxin B-coated beads to selectively bind endotoxin. In humans with sepsis, the use of hemoperfusion columns with polymyxin B (Toraymyxin) has been linked to decreased mortality in cases of septic shock [58]. Permeable membrane devices are newer to the market and remove cytokines and endotoxins by binding to a membrane. These devices also offer some of the same benefits of conventional dialysis, such as the removal of creatine and uremic toxins, and clinical trials determining efficacy are ongoing.

While cytokine removal via hemadsorption shows early promise as one component of therapy in septic patients across species, more research, including blinded, randomized, placebo-controlled multicenter clinical trials, is needed to optimize technique and patient selection to maximize efficacy and reduce potentially deleterious effects [67,68].

## 4. Conclusions

Sepsis is a leading cause of mortality and high hospitalization costs in both human and equine patients. Despite recent attention in the literature to novel treatments for sepsis in both species, supportive care is still the mainstay of treatment. Although cytokine targeting holds promise as a potential treatment strategy for sepsis, significant challenges remain, such as individual patient immune responses, identifying patients likely to benefit from therapy, and translating preclinical findings into effective therapies for septic patients. A better agreement on the definition of sepsis within the equine community could ultimately help to move clinical trials involving horses forward. This lack of consensus definition often complicates inclusion criteria for sepsis-driven studies and presents a challenge in determining the most at-risk populations. The expense of mAb therapy is often cost-prohibitive due to the volume required in the treatment of horses, which limits their use as a novel therapy in equine patients. As hemoperfusion carries a much lower cost, it may represent a feasible novel treatment in horse populations with sepsis.

The ideal degree of cytokine removal in sepsis patients, to achieve balance between the host defense against pathogens and the mitigation of organ damage, coagulopathies, and immune-mediated pathology, is unknown. Further research is needed to better understand the underlying mechanisms of cytokine dysregulation in sepsis and to optimize strategies for cytokine modulation in different patient populations.

## Figures and Tables

**Figure 1 cells-13-01489-f001:**
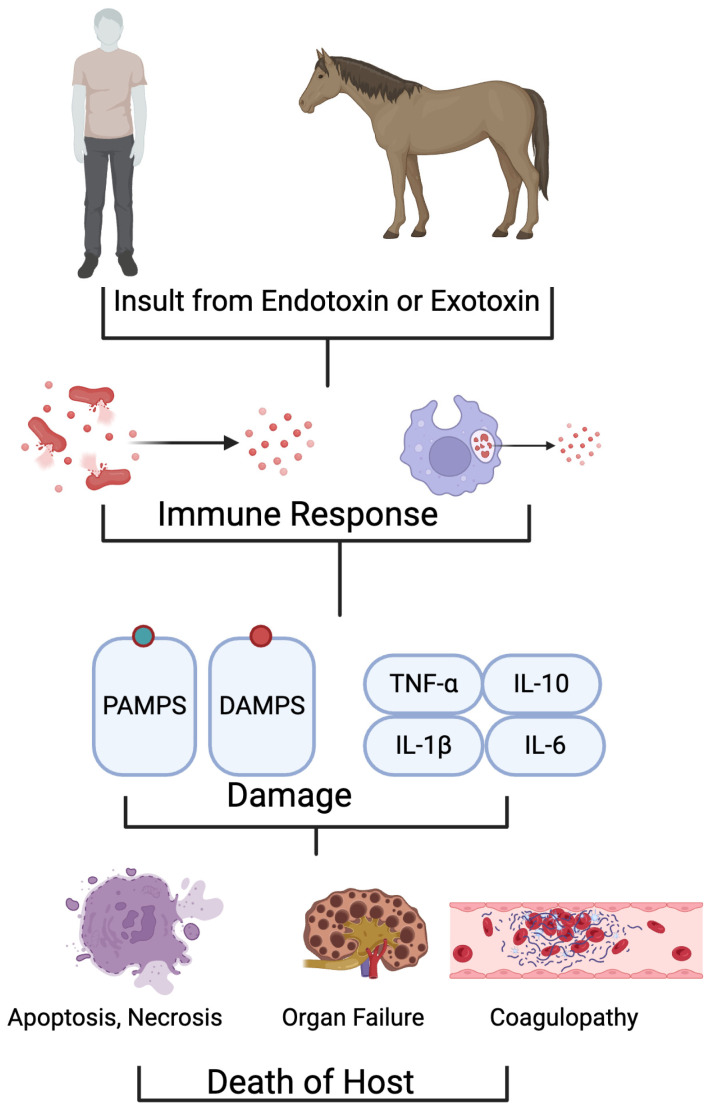
Shared pathology between humans and horses during insult from endotoxins or exotoxins during sepsis. This insult results in the production of PAMPS, DAMPS, and cytokines, leading to downstream effects such as cell death, organ failure, coagulopathies, and host death in both species.

**Figure 2 cells-13-01489-f002:**
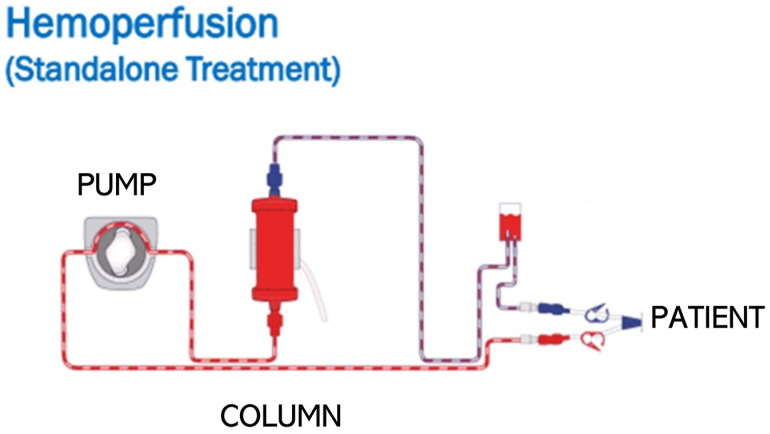
Illustration of hemoperfusion, where blood is removed from the patient (purple), filtered through a column, and returned to the patient (red). Image used with permission from Cytosorb USA^®^.

**Table 1 cells-13-01489-t001:** Selected manuscripts in both species relating to cytokine concentrations in cases of sepsis.

Species	Cytokine	Author/Year	Paper	Major Finding
Human	IL-1β	Jimenez-Sousa, 2017	IL-1B rs16944 polymorphism is related to septic shock and death	Frequency of septic shock higher in patients with IL-1β rs16944 genotype.
	TNF-α	Debets, 1989	Plasma tumor necrosis factor and mortality in critically ill septic patients	Sepsis is accompanied by detectable circulating TNF in 25% of the cases, and for these patients mortality is twice that for comparable TNF-negative patients.
Human	TNF-α, IL-1	Damas, 1989	Tumor necrosis factor and interleukin-1 serum levels during severe sepsis in humans	There was a correlation between the TNF alpha level and sepsis severity score as well as with mortality. In contrast, IL-1 beta serum levels were only slightly increased and were not correlated with severity or mortality.
Human	IL-6, IL-10	Kellum, 2007	Understanding the Inflammatory Cytokine Response in Pneumonia and Sepsis Results of the Genetic and Inflammatory Markers of Sepsis (GenIMS) Study	Highest risk of death was with combined high levels of IL-6 and IL-10.
Human	IL-10	Sherry RM, 1996	Interleukin-10 is associated with the development of sepsis in trauma patients	Plasma IL-10 concentrations were associated with development of hypotension and sepsis.
Human	IL-6	Vivas, 2021	Plasma interleukin-6 levels correlate with survival in patients with bacterial sepsis and septic shock	Patients who developed septic shock maintained high concentrations of IL-6 and had lower survival to those that maintained low IL-6 concentrations.
Human	IL-6, IL-10	Zhang, 2022	Evaluating IL-6 and IL-10 as rapid diagnostic tools for Gram-negative bacteria and as disease severity predictors in pediatric sepsis patients in the intensive care unit	IL-6 and IL-10 are comparably effective in discriminating Gram+ or Gram- sepsis in pediatric intensive care unit (PICU) patients.
Human	IL-1β, IL-6,	Mera, 2011	Multiplex cytokine profiling in patients with sepsis	IL-1β, IL-6, IL-8, IL-12, interferon-γ, granulocyte colony-stimulating factor, and tumor necrosis factor-α exhibited persistent increases in non-survivors.
Human	IL-6, IL-10	Pin-Wu, 2009	Serial cytokine levels in patients with severe sepsis	IL-6 and IL-10 were the key cytokines in the pathogenesis of severe sepsis. IL-6 was comparatively more associated with septic shock and IL-10 was comparatively more associated with mortality.
Equine	TNF-α	Morris, 1991	Tumor necrosis factor activity in serum from neonatal foals with presumed septicemia	Increased serum TNF-a is associated with disease severity in septic foals.
Equine	TNF-α	MacKay, 1991	Tumor necrosis factor activity in the circulation of horses given endotoxin	TNF activity is elevated in horses with experimental endotoxemia and is associated with clinical signs.
Equine	TNF-α	Morris, 1991	Serum tumor necrosis factor activity in horses with colic attributable to gastrointestinal tract disease	Possible association between colic and serum TNF activity in horses and that high mortality may be associated with horses with markedly increased serum TNF activity.
Equine	IL-10, TNF-α, IL-1β	Pusterla, 2006	Expression of molecular markers in blood of neonatal foals with sepsis	Cytokine profiles in septic foals may suggest an immunosuppressive state.
Equine	TNF-α,IL-1β,IL-10	Anderson, 2020	Depletion of pulmonary intravascular macrophages rescues inflammation-induced delayed neutrophil apoptosis in horses	Apoptosis was delayed in horses with elevated TNF-a in SIRS.
Equine	IL-1β, IL-8, IFN-g	Castagnetti, 2011	Expression of interleukin-1β, interleukin-8, and interferon-γ in blood samples obtained from healthy and sick neonatal foals	Evaluation of IL-1β expression in sick neonatal foals can help identify those with sepsis.
Equine	IL-6, IL-10	Burton, 2009	Serum interleukin-6 (IL-6) and IL-10 concentrations in normal and septic neonatal foal	Serum IL-6: IL-10 ratio is likely to provide a valuable prognosticator for neonatal septicemia.

**Table 2 cells-13-01489-t002:** Mechanism of action and available targeted therapies of four major cytokines that are the focus of primary sepsis literature.

Cytokine	Cell Types	Function in Sepsis	Therapies that Target
IL-1β	Macrophages, Monocytes	Apoptosis, Cell Proliferation, Differentiation	Steroids, Hemoperfusion, Monoclonal Antibodies
IL-6	T-cells, Macrophages, Endothelial Cells	Apoptosis, Cell Proliferation, Differentiation, Cytokine Production	Steroids, Hemoperfusion, Monoclonal Antibodies, Cytokine Antagonist
TNF- α	Macrophages,CD4 T-Cells, NK Cells	Cell Proliferation, Cytokine Production, Apoptosis, Tumor Necrosis	Steroids, Hemoperfusion, Monoclonal Antibodies, Cytokine Antagonist
IL-10	Th2 Cells, B-Cells, Monocytes	Inhibition of Inflammatory Cytokines	Steroids, Hemoperfusion, No Other Approved Therapy

## Data Availability

Not applicable.

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
