# Peer review of "A Comparative Review of Cytokines and Cytokine Targeting in Sepsis: From Humans to Horses"

_cells, 2024, doi:10.3390/cells13171489_

Round 1

Reviewer 1 Report

Comments and Suggestions for Authors

The author summarizes in this review, the current status of “shared cytokines and cytokine targeting in sepsis from humans to horses”. The paper also briefly describes an overview of historical and updated perspectives on cytokine targeting therapy in sepsis. However, the manuscript did not explain importance understanding the other major cytokines’ s role of sepsis with developing therapeutic targets. For reader’s understanding, I could not find any tables and graphical data explaining the role and connection of cytokines between human sepsis and horses’ sepsis as well as updated referencing. I would suggest a major revision.

Author Response

Thank you for your comments! The authors agree that the manuscript could benefit from the addition of figures and tables as a result Figure 1 and Figure 2 as well as Table 1 and Table 2 have been added to the manuscript. The authors additionally agree that more information on the link between humans and horses and some of the treatment options could be expanded. As a result the literature surrounding therapy has been further reviewed and a summary of ongoing clinical trial and papers that support the link between the two species has been added. We highly appreciate your time and consideration of this manuscript.

Reviewer 2 Report

Comments and Suggestions for Authors

The paper titled "Comparing Cytokines and Their Impact, on Sepsis; Insights from Humans and Horses" offers an analysis of the similarities between sepsis in humans and horses focusing on how cytokines behave and potential treatment approaches. The authors present a thought-provoking comparison that highlights the value of using equine models to advance our understanding and management of sepsis.

The introduction section effectively outlines how both humans and horses are vulnerable to endotoxins and exotoxins which can trigger sepsis and its complications. The authors underscore the high mortality rates in both species due to sepsis emphasizing the impact well. The introduction is structured well providing a rationale for utilizing horses as a model for sepsis based on similarities in immune responses and clinical features.

The part discussing the pathophysiology of sepsis is thorough and educational detailing the dysregulated response that defines this condition. The authors explain the shift from a stage to immune suppression termed immune paralysis. This detailed overview is crucial, for grasping the complexity of sepsis management challenges.

The discussions on sepsis pathophysiology covering current perspectives sets the stage for an exploration of cytokines. The conclusions effectively emphasize the takeaways of the study underscoring the impact of sepsis in terms of mortality rates and financial burdens on both human and equine populations. The authors aptly emphasize the potential of cytokine targeting as a treatment approach while acknowledging existing challenges like variations in responses and translating lab findings into therapies. The statement for further research is justified, stressing the importance of investigations into cytokine modulation to enhance outcomes for sepsis patients, across different species.

 I consider this review is a significant contribution to the field.

I therefore recommend that the paper can be published in current form.

Author Response

Thank you so much taking for your kind comments regarding our manuscript! 

Reviewer 3 Report

Comments and Suggestions for Authors

The authors of the review paper “A Comparative Review of Cytokines and Cytokine targeting in Sepsis: From Humans to Horses have reviewed shared cytokine physiology across humans and horses in acute sepsis as well as updated views on cytokine targeting therapy. The potential benefits of increased knowledge of equine cytokine mechanisms were discussed and the authors pointed out that horses may serve as a good model for human sepsis, positively impacting our understanding of human sepsis and its treatment.

The topic is relevant and interesting for broad audience. However, the review is very short and should be more informative. More details, references and guidance from the authors throughout the paper is needed, with their stands and comments. Careful reading of the submitted manuscript is needed before submitting it.

Comments:

1.      This is rather Mini Review than comprehensive Review paper.

2.      Some illustration and schematic comparing human and equine cytokine levels and their role in sepsis would be very helpful.

3.      Clinical studies targeting cytokines, done so far, should be mentioned and presented in the Table (there are many of them, some even in phase 2/3 of clinical trials). Comparative Table for human and equine sepsis treatment, related to the cytokines should be added, as it would improve clarity and quality of the review paper.

4.      Conclusions of each section should be expanded and include short statement, i.e. author opinion of current knowledge of specific cytokine target in sepsis. Also, overall conclusion should be more specific for review topic.

5.      Gaps in the knowledge related to the sepsis treatment and promising future therapeutic agents or procedures should be highlighted and some potential strategies depicted. 

6.      In the section 2.2. is written: “For example, 124 horses with abdominal pain that have higher peritoneal fluid IL-6 concentrations at admission are more likely 125 to develop complications such as laminitis or require euthanasia due to poor prognosis. Additionally 126 increased serum concentrations of IL-6 in horses have been noted in mares with placentitis and in horses 127 with equine metabolic syndrome.” I have NO idea what numbers 124, 125, 126, and 127 mean??? Are they some internal references??? Please check and clarify??? Similar non-sense numbers are present also further in the text (raw 193, number 201 appears, etc.).

7.      Raw 96: 33 documented cytokines in sepsis: in both humans and horses??? What are most significant differences other than 4 cytokines that are most important in both species???

8.      List of abbreviations will be very helpful for the readers.

Comments on the Quality of English Language

English is good, should be moderately checked throughout of the paper.

Author Response

  1. This is rather Mini Review than comprehensive Review paper.

Thank you for pointing this out. The authors have added additional information, schematics and tables to the document. The authors are happy to even go more in depth, if necessary, but as sepsis has been well reviewed in literature and cytokine pathology is just emerging, we wanted to keep this review as succinct as possible.

  1. Some illustration and schematic comparing human and equine cytokine levels and their role in sepsis would be very helpful.

      Thank you for pointing this out we have added an illustration about the shared pathology (Figure 1). Unfortunately very little is known in either species about cut offs for concentrations and the role in sepsis pathology, which we hope this review will spark further interest in elucidating these interesting mechanisms. We have added a Table 1 which highlights recent clinical literature about cytokine concentrations in sepsis.

  1. Clinical studies targeting cytokines, done so far, should be mentioned and presented in the Table (there are many of them, some even in phase 2/3 of clinical trials). Comparative Table for human and equine sepsis treatment, related to the cytokines should be added, as it would improve clarity and quality of the review paper.

Thank you for pointing this out. A paragraph about clinical studies has been added. As there are currently 96 ongoing clinical studies just in humans and no database for equine clinical trials the authors feel this would be a very extensive table to add. A comparative table (Table 2) has been added to the manuscript and well as Table 1 that highlights clinical studies.

  1. Conclusions of each section should be expanded and include short statement, i.e. author opinion of current knowledge of specific cytokine target in sepsis. Also, overall conclusion should be more specific for review topic.

Thank you for pointing this out. The conclusion has been modified to include the author opinion of current knowledge and further topic specific comments.

  1. Gaps in the knowledge related to the sepsis treatment and promising future therapeutic agents or procedures should be highlighted and some potential strategies depicted.

Thank you for pointing this out. A paragraph about gaps in knowledge has been added to the conclusion. Also potential pitfalls of each treatment have been added to each of the treatment paragraphs.

  1. In the section 2.2. is written: “For example, 124 horses with abdominal pain that have higher peritoneal fluid IL-6 concentrations at admission are more likely 125 to develop complications such as laminitis or require euthanasia due to poor prognosis. Additionally 126 increased serum concentrations of IL-6 in horses have been noted in mares with placentitis and in horses 127 with equine metabolic syndrome.” I have NO idea what numbers 124, 125, 126, and 127 mean??? Are they some internal references??? Please check and clarify??? Similar non-sense numbers are present also further in the text (raw 193, number 201 appears, etc.).

Thank you for pointing this out! The journal transferred the template after we submitted, these appear to be the old-line numbers from our original submission. We have removed them!

  1. Raw 96: 33 documented cytokines in sepsis: in both humans and horses??? What are most significant differences other than 4 cytokines that are most important in both species???

Thank you for pointing this out. The authors agree this is confusing and have removed the total of documented cytokines and expanded on why the inclusion of the primary 4 listed in this article was chosen.

  1. List of abbreviations will be very helpful for the readers.

      Thank you for pointing this out. We have added a list of abbreviations!

Round 2

Reviewer 1 Report

Comments and Suggestions for Authors

I am glad that the authors modified the manuscript accordingly. The current version of the manuscript is quite clear, precise and easy to understand. Also, the authors included summary of the tables and figures. So, the present format of the manuscript is acceptable to publish. 

Reviewer 3 Report

Comments and Suggestions for Authors

The authors have answered all my questions and concerns and improved the paper considerably. I have no further comments or questions for the authors.

Comments on the Quality of English Language

Minor editing of English is required!